# Association of skirt size and postmenopausal breast cancer risk in older women: a cohort study within the UK Collaborative Trial of Ovarian Cancer Screening (UKCTOCS)

Evangelia-Ourania Fourkala,[1] Matthew Burnell,[1] Catherine Cox,[1] Andy Ryan,[1] Laura Currin Salter,[2] Aleksandra Gentry-Maharaj,[1] Mahesh Parmar,[3] Ian Jacobs,[1,4] Usha Menon[1]

► Prepublication history and additional material is available. To view please visit the journal (http://dx.doi.org/10.1136/bmjopen-2014-005400).

E-OF and MB are joint first author.

For numbered affiliations see end of article.

**Correspondence to**
Professor Usha Menon;
u.menon@ucl.ac.uk

## ABSTRACT

**Objectives:** Several studies suggest that overall and central-obesity are associated with increased breast cancer (BC) risk in postmenopausal-women. However, there are no studies investigating changes of central obesity and BC. We report on the association of BC risk with self-reported skirt size (SS; waist-circumference proxy) changes between 20s and postmenopausal-age.

**Design:** Prospective cohort-study.

**Setting:** UK Collaborative Trial of Ovarian Cancer Screening (UKCTOCS) involving the nine trial centres in England.

**Participants:** Postmenopausal-women aged >50 with no known history of BC prior to or on the day of completion of the study-entry questionnaire.

**Interventions:** At recruitment and at study entry, women were asked to complete a questionnaire. Women were followed-up via 'flagging' at the NHS Information Centre in England and the Hospital Episode Statistics.

**Main outcome-measure:** Time to initial BC diagnosis.

**Results:** Between 2 January 2005 and 1 July 2010, 92 834 UKCTOCS participants (median age 64.0) completed the study-entry questionnaire. During median follow-up of 3.19 years (25th–75th centile: 2.46–3.78), 1090 women developed BC. Model adjusted analysis for potential confounders showed body mass index (BMI) at recruitment to UKCTOCS (HR for a 5 unit change=1.076, 95% CI 1.012 to 1.136), current SS at study entry (HR=1.051; 95% CI 1.014 to 1.089) and change in SS per 10 years (CSS) (HR=1.330; 95% CI 1.121 to 1.579) were associated with increased BC risk but not SS at 25 (HR=1.006; 95% CI 0.958 to 1.056). CSS was the most predictive singe adiposity measure and further analysis including both CSS and BMI in the model revealed CSS remained significant (HR=1.266; 95% CI 1.041 to 1.538) but not BMI (HR=1.037; 95% CI 0.970 to 1.109).

**Conclusions:** CSS is associated with BC risk independent of BMI. A unit increase in UK SS (eg, 12–14)

## Strengths and limitations of this study

- To the best of our knowledge, this is the first study investigating the association between central obesity using skirt size (SS) as a proxy and breast cancer risk. Between 25 and postmenopausal age, an increase in SS by one unit every decade increased the risk of postmenopausal breast cancer by 33% while decrease in SS was associated with lowering of risk.
- Our prospective cohort-study includes 94 000 women with comprehensive follow-up through data linkage to multiple national databases.
- There is a possibility of underestimation of self-reported SS. However, if current SS at study entry is uniformly underestimated then there is merely rescaling of CSS so that the strength of the association is unaffected. Furthermore, recall bias of the SS at 25 maybe a limitation but unless this inability in reporting is systematically related to future breast cancer, measurement error can only result in underestimating the strength of the true association between CSS and breast cancer risk.
- Given that obesity is now emerging as a global epidemic, from a public health prospective these findings are significant as they provide women with a simple and easy to understand message.

every 10-years between 25 and postmenopausal-age is associated with postmenopausal BC risk by 33%. Validation of these results could provide women with a simple and easy to understand message.

**Trial registration number:** ISRCTN22488978.

## INTRODUCTION

Breast cancer is a major global health concern with 1 384 000 women being diagnosed every year.[1] Significant advances in

breast cancer prevention have been accompanied by growing controversies regarding breast cancer screening. Both highlight the need for better risk stratification. During the past few years a growing number of studies have reported on anthropometric factors, particularly on the relationship between breast cancer risk and overall and/or central obesity.[2–4]

A number of adiposity measures exist. The oldest and most common measure is body mass index (BMI) which is an empirical proxy for human body fat based on an individual's weight and height. Central fat is another measure of obesity. It is usually estimated by waist circumference, waist–hip ratio, waist–height ratio or the conicity index which evaluates waist circumference in relation to height and weight. An adequate and easy to use proxy measure for waist circumference is clothing (trouser/skirt) size, which has been shown to provide a reliable and feasible estimate of waist circumference at the population level.[5] It has been suggested that waist circumference independent of BMI, maybe predictive of increased health risk.[6]

There is a large body of evidence that high BMI is associated with increased breast cancer risk in postmenopausal women and reduced risk in premenopausal women.[3 4 7] Another suggested breast cancer risk predictor is adult weight gain, which typically reflects an increase in body fat.[8–10] Results concerning weight loss are more equivocal; there are studies that show no significant effect on risk, while some suggest postmenopausal weight loss may be protective.[8 11–14] Furthermore, the association of central obesity with breast cancer risk is unclear with studies reporting conflicting results.[2 15] Among postmenopausal women, several studies[16–23] but not all[13 24–28] have shown waist circumference or waist-to-hip ratio to be positively associated with increased breast cancer risk.[29] There are, however, no studies examining changes in waist circumference or waist-to-hip ratio and breast cancer risk.

The UK Collaborative Trial of Ovarian Cancer Screening (UKCTOCS),[30] with comprehensive follow-up, provides the opportunity to examine the association of body size changes with breast cancer risk in a large prospective cohort. Furthermore, the availability of socioeconomic, reproductive, lifestyle and cancer family history data makes it possible to adjust for many potential confounders of breast risk.

## METHODS
### Setting
UKCTOCS is a multicentre randomised controlled trial of ovarian cancer screening in the general population. The trial invited 1.2 million women of whom 202 638 were recruited between 17 April 2001 and 27 September 2005 through 13 trial centres in England, Wales and Northern Ireland. The detailed trial design and eligibility criteria have been described elsewhere.[30] All participants completed a questionnaire at recruitment and a further postal follow-up questionnaire 3–4 years post-recruitment. The latter was the study-entry questionnaire for the current analysis. Follow-up was through a 'flagging study' with the Health and Social Care National Health Service (NHS) Information Centre who provided regular notifications of cancer registrations or deaths in the cohort. In addition, data linkage to the Hospital Episode Statistics (HES) an administrative database in England provided information on all inpatient and outpatient NHS hospital visits.

Women provided written consent for use of their data in secondary studies. Ethical approval was obtained from the Joint University College London (UCL)/UCL Hospital Committees on the Ethics of Human Research (REC reference:06/Q0505/102, June 2008).

### Subjects
Participants recruited from England who returned the study-entry questionnaire prior to 2 July 2010 (censorship date—see 'breast cancer notification'). Women recruited from Northern Ireland and Wales were excluded as HES was only available for women residing in England.

### Data collection
#### Questionnaires
*Recruitment questionnaire*: This was completed at recruitment into UKCTOCS and included information on the variables listed in tables 1 and 2. This included height and weight used to calculate BMI, the postcode which

**Table 1** Baseline characteristics and HRs for breast cancer for adiposity measures and potential confounders (continuous variables)

| Factor | Mean | SD | 25th centile | 50th centile | 75th centile | HR | Lower 95% CI | Upper 95% CI | p Value |
|---|---|---|---|---|---|---|---|---|---|
| Skirt size at age 25 years | 12.509 | 2.533 | 12 | 12 | 14 | 1.008 | 0.962 | 1.056 | 0.736 |
| Skirt size at study entry | 15.133 | 3.452 | 12 | 14 | 16 | 1.045 | **1.011** | **1.081** | **0.01** |
| CSS | 0.334 | 0.359 | 0.191 | 0.287 | 0.533 | 1.283 | **1.09** | **1.51** | **0.003** |
| BMI | 26.449 | 4.783 | 23.203 | 25.598 | 28.821 | 1.061 | **1.006** | **1.119** | **0.029** |
| IMD | 0.011 | 1.01 | −0.685 | −0.327 | 0.458 | 1.008 | 0.952 | 1.068 | 0.778 |
| Age at last period | 48.975 | 6.009 | 46.264 | 49.977 | 52.703 | 1.024 | **1.014** | **1.035** | **<0.001** |
| Age at first period | 12.959 | 1.607 | 12 | 13 | 14 | 1.02 | 0.983 | 1.06 | 0.293 |

BMI, body mass index at recruitment (5 kg/m$^2$); CSS, change of skirt size per 10 years.

**Table 2** Baseline characteristics and HRs for breast cancer for potential confounders (categorical variables)

| Categorical variable | All women N (%) | HR | Lower 95% CI | Upper 95% CI | p Value |
|---|---|---|---|---|---|
| **Ethnicity** | | | | | |
| White | 90 011 (97.35) | ref | | | |
| Black | 832 (0.90) | 1.307 | 0.622 | 2.75 | 0.48 |
| Asian | 490 (0.53) | 0.827 | 0.443 | 1.541 | 0.549 |
| Chinese | 166 (0.18) | 0.554 | 0.078 | 3.936 | 0.555 |
| Other | 669 (0.72) | 0.693 | 0.288 | 1.668 | 0.413 |
| No O-level | 56 856 (61.24) | ref | | | |
| O-level | 35 986 (38.76) | 1.208 | 1.07 | 1.363 | 0.002 |
| No A-level | 78 436 (84.48) | ref | | | |
| A-level | 14 406 (15.52) | 1.204 | 1.03 | 1.407 | 0.02 |
| No Clerical or commercial qualification | 67 075 (72.25) | ref | | | |
| Clerical or commercial qualification | 25 767 (27.75) | 1.085 | 0.953 | 1.236 | 0.219 |
| **Education** | | | | | |
| No nursing or teaching | 79 668 (85.81) | ref | | | |
| Nursing or teaching | 13 174 (14.19) | 1.051 | 0.888 | 1.243 | 0.565 |
| No degree college/university | 74 640 (80.39) | ref | | | |
| Degree college/university | 18 202 (19.61) | 1 | 0.859 | 1.163 | 0.996 |
| No other than above qualification | 64 090 (69.03) | ref | | | |
| Other than above qualification | 28 752 (30.97) | 0.828 | 0.724 | 0.947 | 0.006 |
| **Hysterectomy with ovarian conservation at recruitment** | | | | | |
| No | 76 318 (82.20) | ref | | | |
| Yes | 16 524 (17.80) | 0.961 | 0.821 | 1.125 | 0.621 |
| **Oral contraceptive pill use** | | | | | |
| No | 37 083 (39.94) | ref | | | |
| Yes | 55 759 (60.06) | 0.97 | 0.852 | 1.106 | 0.654 |
| **Hormone replacement therapy use at recruitment** | | | | | |
| No | 73 544 (79.21) | ref | | | |
| Yes | 19 298 (20.79) | 1.12 | 0.971 | 1.289 | 0.118 |
| **Hormone replacement therapy use at study entry** | | | | | |
| No | 85 866 (92.50) | ref | | | |
| Yes | 6963 (7.50) | 1.232 | **1** | **1.517** | **0.049** |
| **Sterilisation** | | | | | |
| No | 74 550 (80.30) | ref | | | |
| Yes | 18 292 (19.70) | 0.951 | 0.817 | 1.108 | 0.523 |
| **Infertility treatment** | | | | | |
| No | 72 731 (78.34) | ref | | | |
| Yes | 20 111 (21.66) | 1.459 | **1.103** | **1.93** | **0.008** |
| **Pregnancies less than 6 months** | | | | | |
| 0 | 64 126 (69,90) | ref | | | |
| 1 | 18 896 (20.60) | 1.124 | 0.971 | 1.302 | 0.116 |
| 2 | 5596 (6.10) | 0.966 | 0.742 | 1.256 | 0.795 |
| 3 | 1790 (1.95) | 1.521 | **1.055** | **2.192** | **0.024** |
| 4 | 649 (0.71) | 1.690 | 0.955 | 2.990 | 0.071 |
| 5 | 678 (0.74) | 0.957 | 0.454 | 2.014 | 0.907 |
| **Pregnancies more than 6 months** | | | | | |
| 0 | 10 986 (11.87) | ref | | | |
| 1 | 11 297 (12.18) | 1.023 | 0.821 | 1.276 | 0.837 |
| 2 | 40 890 (44.17) | 0.768 | **0.640** | **0.923** | **0.005** |
| 3 | 20 007 (21.61) | 0.767 | **0.624** | **0.944** | **0.012** |
| 4 | 6612 (7.14) | 0.824 | 0.626 | 1.084 | 0.625 |
| 5 | 2804 (3.03) | 0.594 | 0.386 | 0.915 | 0.385 |
| **Alcohol (units per week) at study entry** | | | | | |
| None | 21 086 (22.97) | | | | |
| Less than 1 | 16 084 (17.52) | 1.011 | 0.835 | 1.226 | 0.905 |
| 1–3 | 18 717 (20.39) | 1.062 | 0.886 | 1.275 | 0.511 |
| 4–6 | 13 813 (15.05) | 1.039 | 0.851 | 1.267 | 0.71 |
| 7–10 | 11 056 (12.04) | 0.949 | 0.762 | 1.181 | 0.64 |

**Table 2** Continued

| Categorical variable | All women N (%) | HR | Lower 95% CI | Upper 95% CI | p Value |
|---|---|---|---|---|---|
| 11–15 | 6549 (7.13) | 0.985 | 0.758 | 1.279 | 0.91 |
| 16–30 | 2990 (3.26) | 1.271 | 0.92 | 1.745 | 0.146 |
| 21+ | 1501 (1.64) | 1.331 | 0.866 | 2.044 | 0.192 |
| Smoking | | | | | |
| 0 | 51 470 (56.27) | ref | | | |
| 250 | 21 661 (23.68) | 0.958 | 0.825 | 1.113 | 0.574 |
| 500 | 9632 (10.53) | 1.032 | 0.847 | 1.259 | 0.752 |
| 750 | 8706 (9.52) | 1.153 | 0.947 | 1.403 | 0.156 |
| Relatives breast cancer history | | | | | |
| No | 72, 731 (78.34) | ref | | | |
| Yes | 20 111 (21.66) | 1.48 | **1.298** | **1.687** | **<0.0005** |
| Relatives ovarian cancer history | | | | | |
| No | 88 878 (95.73) | ref | | | |
| Yes | 3964 (4.27) | 0.946 | 0.699 | 1.279 | 0.718 |

was used to derive the Index of Multiple Deprivation (IMD; with lower scores indicating less deprived areas), age at first and last period, number of pregnancies, hysterectomy, sterilisation, infertility and breast and ovarian cancer family history, past oral contraceptive pill use, current hormone replacement therapy (HRT) use.

*Study-entry questionnaire*: This was completed at follow-up (3–4 years after recruitment) and provided information on education, skirt size, continuing use of HRT, smoking, alcohol use, health status and cancer diagnosis postrecruitment (tables 1 and 2). The date of completion of this questionnaire was the entry date for the current study. This was available in 89.8% of the women. Where this date was missing, the date of questionnaire data entry into the trial management system was used.

### Breast cancer notification

The ICD-9 and ICD-10 codes used to identify women with breast cancer were 174* and C50*, respectively. Up-to-date data records from the NHS Information Centre and HES were searched to identify women with a breast cancer diagnosis. Women also self-reported breast cancer. Both at recruitment and at current study entry, women were asked specifically whether they had ever been diagnosed with breast cancer and if so, details of diagnosis date and details of treating physician.[31] Overall there was a good concordance between the different sources used to identify the breast cancer cases.[31 32]

As there can be delays in cancer notification by the National Health and Social Care NHS Information Centre,[31] the censorship date was set as 1 July 2010, 2 years prior to the date (10 July 2012) when data was provided by the NHS Information Centre agencies for this analysis. This ensured that as far as possible, all breast cancers in the stated period of observation were registered and therefore included. HES data was available for all the participants for the period between 6 August 1998 when HES started and 31 March 2010.

### Exposure variables

Self-reported height and weight were taken from the recruitment questionnaire and used to calculate BMI (with one unit equal here to $5 \, \text{kg/m}^2$). In the study-entry questionnaire two specific questions related to skirt size (SS) were asked: (1) 'What was your SS when you were in your twenties?' (2) 'What is your SS now?' Women could choose from 13 SS categories ranging from 6 to 30. The two SS questions were used to create a variable reflecting change in SS over time. The change in SS variable (CSS) was calculated as an increase in SS per 10 years. Note that a 'one unit' increase in SS means an increase by two nominal values, for example from size 10 to size 12 as odd sizes do not exist in the UK.

### Statistical analysis

Standard survival analysis methods were used to analyse the data. Cox models were preferred so that specification of an underlying hazard was not required. Women with breast cancer previous to questionnaire completion date were excluded from the analysis. The date of study-entry questionnaire was the time point used for entry into study. However, there was delayed entry (left truncation) in that they were only able to participate in the current study because they 'survived' (had not died, withdrawn from trial or got breast cancer) the period from their mid-20s (the first skirt size question) to study-entry questionnaire completion date (the second current skirt size question). To account for a woman's contribution to the analysis being conditional on having 'survived' during the unobserved period, the length of which depends on the age at study entry, the timescale used was age rather than date. Therefore, the time origin was fixed at age=25 for all participants and study entry was age at completion of study-entry questionnaire. Women were censored either at age at first breast cancer, death from any cause or 1 July 2010 whichever occurred first.

Summaries of all identified potential confounding risk factors were performed. These variables were also included individually in a Cox regression model to obtain univariable estimates of their HR relating to breast cancer risk. In addition kernel density plots of all four adiposity measures were created and Pearson correlation coefficients were calculated for all adiposity measure-pairings. To obtain an appropriate confounder-adjusted estimate of the relative hazard of breast cancer of increasing SS, a Cox regression model including CSS in continuous form with all potential risk factors, regardless of statistical significance, was fitted (the 'full model'). The functional form for CSS was determined using fractional polynomials, by considering powers from the standard predefined set {−2, −1, −0.5, 0, 0.5, 1, 2, 3} and modelling the most appropriate relationship between CSS and the (log) relative hazard. The full model was refitted with each of the adiposity measures forming the exposure variable, in turn. In addition, a variant formulation of the full model included both CSS and BMI and both SS at current study entry and BMI.

Cox-Snell residuals were used to test the model fit and the proportional hazards assumption used in Cox regression was tested using the scaled Schoenfeld residuals for each variable in the final model. The (undue) influence of any outliers was assessed by calculating the change in the parameter estimate when an observation is deleted. To visualise the 'dose-response' relationship model, predicted relative hazards with 95% confidence bands for a range of CSS values were plotted. To obtain estimates of absolute risk for certain scenarios (when CSS=0, CSS=1, CSS=median value), the event density function was assumed to be exponentially distributed and the full model with CSS was fitted again using exponential regression. Survival curves with the same covariate pattern following both the Cox and exponential model fitting were plotted to ensure that the constant hazard was a reasonable assumption when averaged over the timeframe. An estimate of absolute risk was calculated by using the covariate-specific hazard from the exponential model for 1 and 5 years.

### Sensitivity analyses

To guard against the possibility of reverse causality whereby preclinical breast cancer may result in weight gain, the full model with CSS was refitted but with a time-lag of 1 year so that the start of the study period occurs 1 year after completion of the study-entry questionnaire.

In the primary analysis, missing data in CSS or any other confounder was handled by case-wise deletion. To assess the robustness of estimation with such an approach to potential biases, multiple imputation (MI) was used to calculate the corresponding model estimates with missing data imputed appropriately. Specifically, a multivariate normal distribution formed by all potential predictors and the outcome variables (event and time) was used for the imputed random draws, and 20 fully imputed data sets were created with which to obtain valid estimates and SEs.

## RESULTS

A total of 157 997 women residing in England were recruited to the trial between 17 April 2001 and 29 September 2005. Of 11 659 (9440 withdrew, 533 moved away, 1686 died) women were not sent the study-entry questionnaire. The remaining 146 338 women were sent the study-entry questionnaire between 28 June 2005 and 17 March 2009 and 112 945 (77.2%) responded by 31 December 2010. A total of 10 689 of these women completed the current study-entry questionnaire after the censorship date of 1 July 2010 and were therefore excluded.

Eight women did not give consent for flagging. The remaining 102 249 were successfully flagged with the NHS Information Centre. Of these, 62 were lost to follow-up by the NHS Information Centre prior to questionnaire completion, 9342 were diagnosed with breast cancer before or on the day of completion of the study-entry questionnaire and in 10, study-entry questionnaire completion date could not be calculated. The remaining 92 834 women were included in the study and were followed-up for a median of 3.19 years (25th–75th centile: 2.46–3.78 years). During the study period, 1090 women developed breast cancer, resulting in an absolute risk of breast cancer of 1.2% (1090/92 834) in this cohort of postmenopausal women, 75% of whom were aged over 60 years with no prior history of breast cancer. More information regarding the age distribution of the cohort is provided in online supplementary figure S1.

The reproductive and lifestyle characteristics of the 92 834 women included in the analysis are summarised in tables 1 and 2. Briefly, the women were mainly White, with a median year of birth at 1943 (IQR: 1937–1947), 19.6% had a University degree and their median standardised—IMD (using 2008 data) was −0.327 (IQR: −0.685–0.458) indicating that most women are from less deprived areas. Most women were overweight (median BMI 25.598, (IQR: 23.203–28.821) at trial recruitment (median recruitment age 60.2; IQR: 55.8–65.8), had a median SS at 25 of 12 (IQR: 12–14) and at study entry of 14 (IQR: 12–16) (median age 64.0; IQR: 59.7–69.7). The median CSS was 0.287 (IQR: 0.191–0.533) per 10 years, which is equivalent to an increase of 1 unit in SS (eg, 10–12) in almost 35 years. Figure 1 shows the distribution for the four adiposity measures where CSS is largely symmetrical while the other three measures display some positive skewness.

The Pearson's correlation coefficients for the different adiposity measures are presented in table 3. All of them were relatively high correlated apart from SS at 25 with CSS. Among the univariable Cox regressions, all measures of adiposity had significant HRs at the 5% level, except for SS at 25 (tables 1 and 2). Other significant predictors of breast cancer risk were age at last period, number pregnancies greater than 6 months, infertility treatment, family history of breast cancer and HRT use at current study entry (tables 1 and 2).

A total of 5500 women were not included in the model due to missing values for either CSS or a

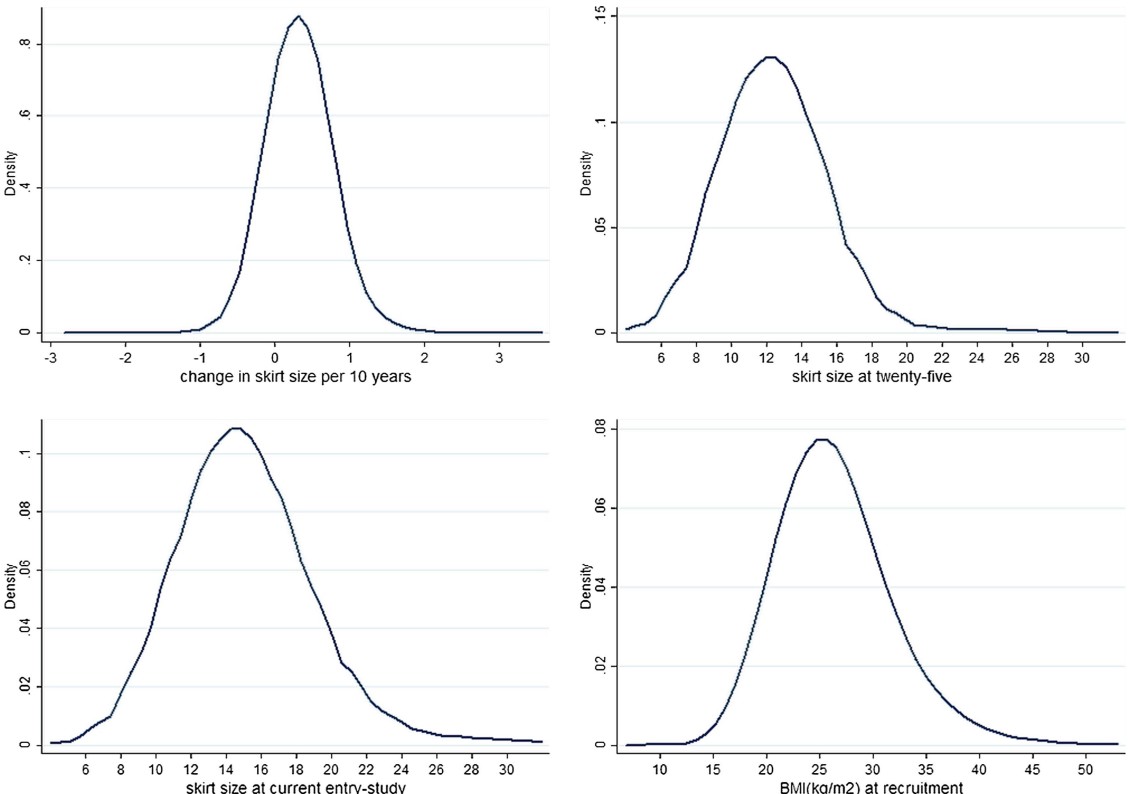

**Figure 1** Relative hazards for a range of skirt size (SS) changes per 10 years (CSS) with 95% confidence bands.

confounder. The remaining 87 334 women, 1013 of whom developed breast cancer during follow-up were included in the full Cox model with CSS. Use of fractional polynomials of the CSS variable showed the most appropriate functional form was linear, with tests of better fit for other powers of degrees 1 (p=0.719) and 2 (p=0.981) showing no improvement. Confounder-adjustment revealed a stronger positive association between CSS and breast cancer risk (HR=1.330 per 10 years; 95% CI 1.121 to 1.579; p=0.001) than without adjustment (table 4). For each unit increase in SS per 10 years the hazard rate of breast cancer increased by 33%. Replacement of CSS with the other adiposity measures in a full model, again showed a significant but lesser association between BMI (HR=1.076 per 5 units; 95% CI 1.018 to 1.136; p=0.009) and SS at study entry (HR=1.051; 95% CI 1.014 to 1.089; p=0.006) with breast cancer risk, but not for SS at 25 (HR=1.006; 95% CI 0.958 to 1.056; p=0.809). Including recruitment BMI into the full model with CSS showed some sharing of predictive power but

CSS had an association with breast cancer risk independent of BMI (HR=1.27; 95% CI 1.041 to 1.538; p=0.018 for CSS; HR=1.04; 95% CI 0.970 to 1.109; p=0.287 for recruitment BMI; table 4). A full model with SS at study entry (HR=1.037; 95% CI 0.992 to 1.085; p=0.109) and BMI (HR=1.037; 95% CI 0.964 to 1.116; p=0.329) suggested less independence between the two measures than between CSS and BMI and neither variable was statistically significant at α=0.05 (table 4).

Tests and plots based on the Schoenfeld residuals showed no evidence of any violation of the proportional hazards assumption. Online supplementary figure S2 plots these residuals for CSS with a superimposed smoother and shows that the effect of CSS on breast cancer risk is essentially stable over the entire age range considered in this study. The Cox-Snell residuals plotted against their cumulative hazard showed a virtual line of unity indicating good model fit (figure not shown). Similar diagnostic results were found using the other adiposity measures.

**Table 3** Pearson correlation coefficients between skirt size at 25, and at study entry (follow-up questionnaire completion), change in skirt size (CSS) per 10 years and body mass index (BMI) at recruitment

|  | Skirt size at 25 | Skirt size at study entry | BMI at recruitment | CSS |
|---|---|---|---|---|
| Skirt size at 25 | 1.000 | | | |
| Skirt size at study entry | 0.606 (p<0.0005) | 1.000 | | |
| BMI | 0.330 (p<0.0005) | 0.643 (p<0.0005) | 1.000 | |
| CSS | −0.153 (p<0.0005) | 0.693 (p<0.0005) | 0.497 (p<0.0005) | 1.000 |

**Table 4** Confounder-adjusted Cox model estimates for various adiposity measures and their association with breast cancer risk

| Measures of adiposity | Effective sample size | Number of events | HR | Lower 95% CI | Upper 95% CI | p Value |
|---|---|---|---|---|---|---|
| Full model with individual adiposity measure | | | | | | |
| CSS | 87 334 | 1013 | 1.33 | 1.121 | 1.579 | 0.001 |
| Skirt size at 25 | 87 783 | 1017 | 1.006 | 0.958 | 1.056 | 0.829 |
| Skirt size at study entry | 87 661 | 1018 | 1.051 | **1.014** | **1.089** | 0.006 |
| BMI at recruitment | 87 615 | 1021 | 1.076 | **1.012** | **1.136** | 0.009 |
| Full model with specific adiposity measure combinations | | | | | | |
| CSS | 86 678 | 1008 | 1.266 | 1.041 | 1.538 | 0.018 |
| BMI | | | 1.037 | 0.97 | 1.109 | 0.287 |
| Skirt size at study entry | 87 003 | 1013 | 1.037 | 0.992 | 1.085 | 0.109 |
| BMI | | | 1.037 | 0.964 | 1.116 | 0.329 |
| Full model with one year time lag | | | | | | |
| CSS | 86 284 | 650 | 1.422 | **1.149** | **1.758** | 0.001 |
| Full model with multiple imputation model | | | | | | |
| CSS | 92 834 | 1090 | 1.3 | 1.102 | 1.534 | 0.002 |

Full model means adjusted for all potential confounders: Index of Multiple Deprivation (IMD), age at first and last period, number of pregnancies, hysterectomy, sterilisation, infertility, breast and ovarian cancer family history, oral contraceptive use (pill) use, hormone replacement therapy (HRT) use (from recruitment questionnaire); education, skirt size, continuing use of HRT, smoking, alcohol use, health status and cancer diagnosis (from study-entry questionnaire).
BMI, body mass index at recruitment per 5 kg/m$^2$; CSS, change of skirt size per 10 years.

Figure 2 shows how the model relates breast cancer risk to varying changes in SS. For example, for those with an increase of 2 SS every 10 years, the estimated relative hazard increases by 77% (HR=1.769; 95% CI 1.164 to 2.375). Online supplementary figure S3 shows the estimated survival functions when CSS=0, CSS=1 and CSS=0.287 (the median CSS value), with other continuous covariates set to their mean value and categorical variables set to their base value and starting at the earliest observed age entering the study. These functions are plotted for both the Cox and exponential model and show that the exponential model is a reasonable approximation over the timeframe considered here. The derived estimates of absolute risk for 1 year were 0.33% (95% CI 0.30% to 0.36%) when CSS=0 and 0.43% (95% CI 0.37% to 0.48%) when CSS=1 (exponential model

HR=1.31). These estimates compare with 0.35% (95% CI 0.33% to 0.38%) when CSS=0.287 (median CSS) and the raw incidence rate of 0.38% (1090 cancers in 3.07 mean study years). The 5 year absolute risks are 1.63% (95% CI 1.48% to 1.77%) when CSS=0 and 2.14% (95% CI 1.87% to 2.40%) when CSS=1 and a raw incidence rate (number of cancers per 5 years of study time across sample) of 1.91%.

A sensitivity analysis added a time-lag of 1 year to the primary analysis that used the full model with CSS. For this regression model only 86 284 women were included as 1 121 women were left-censored before 1 year, including 71 who had missing covariate data. The HR of 1.422 (95% CI 1.149 to 1.758; p=0.001) is slightly larger compared to the primary analysis but not substantively different (table 4).

A multiple imputation analysis was carried out to investigate the effect of missing data on the model estimation. Data were missing for one or more variables in 5.9% and for CSS in 1.2% of the sample. Twenty completed data sets were created using multiple imputation and the subsequent HR=1.300 (95% CI 1.102 to 1.534; p=0.002) (table 4) and SE estimate for CSS was essentially unchanged compared to the case-wise model result.

## DISCUSSION

This is to date the only study that we are aware of investigating the association of change in SS and breast cancer risk. Our data suggests that a unit increase in UK SS (eg, 12–14) every 10 years between the 20s and postmenopausal age increases the risk of postmenopausal breast cancer by 33%. We estimated an increase in 5-year absolute risk of postmenopausal breast cancer from 1 in 61 to 1 in 51 with each unit increase in SS per 10 years.

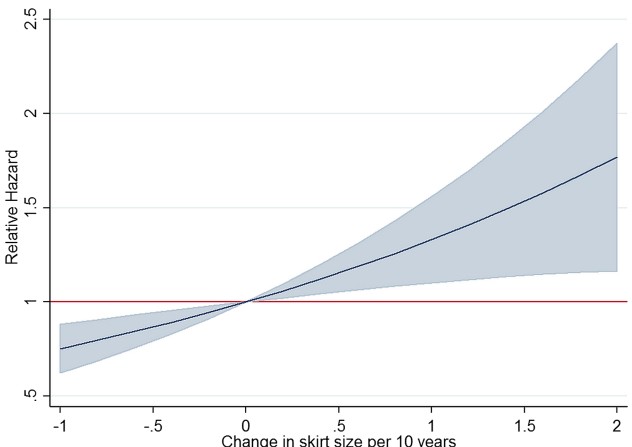

**Figure 2** Distribution for skirt size (SS) at 20 s, skirt size at current entry-study, BMI at recruitment and change of skirt size (CSS) every 10 years.

Reductions in SS decreased the risk of breast cancer (figure 2) though it should be noted that for 76% of the women in this study, SS increased over their adult lifetime. These findings may provide women with a simple and easy to understand message given that SS has been found to be a reliable measure for changes in waist circumference and one that women may relate and understand better in comparison to other adiposity measures such as BMI.[5]

The major strengths of this study are its prospective cohort design, sample size, comprehensive follow-up through national registries and administrative databases and self-reporting resulting in high-quality data. Evaluation of numerous possible confounders including life style choices and socioeconomic status minimised bias. Owing to missing data in the covariates, case-wise deletion meant 5508 women were dropped from the Cox regression model. However, a sensitivity analysis using 20 completed sets of multiply imputed data showed that the missing data had little effect on the overall outcome of the study. Furthermore, the study conclusions were maintained after introducing a time-lag of 1 year to the study entry to rule out reverse causality. The absolute risk of breast cancer in this cohort over a median follow-up of 3.2 years was 1.2% as only postmenopausal women with no prior history of breast cancer, 75% of whom were aged over 60 years, were included in this analysis. In keeping with other studies, family history of breast cancer, infertility, age at last period, HRT use were associated with increased breast cancer risk while pregnancies lasting over 6 months were protective.[33] Limitations of the study are the relatively short follow-up period, the possibility of underestimation of self-reported SS and recall bias of SS at 25. However, if current SS is uniformly underestimated across the sample then there is merely a rescaling of the CSS variable so that the strength of the relationship is unaffected, although the HR relating to a unit increase would be smaller. The same scenario applies to possible modern downsizing in SS to reflect increasing BMI (eg, a SS of 12 in their mid-20s is now equivalent to, say, a SS of 10) assuming that this downsizing trend does not noticeably occur over the period of 'current study-entry SS'. All dates of FU questionnaire were between 2005 and 2010 with 97% between 2006 and 2009. The true relationship would only be weaker than reported if underestimation was not random but more prevalent among those who got breast cancer compared to those that did. Furthermore, unless the inability to accurately recall SS in the 20 s is systematically related to future breast cancer or a confounder, measurement error can only result in underestimating the strength of the true relationship between CSS and risk of breast cancer.

The findings that CSS is associated with breast cancer risk (CSS per 10 years HR=1.33, 95% CI 1.12 to 1.58; p=0.0010) and that addition of BMI to the model does not significantly improve risk prediction, are consistent with previous studies addressing the relationship between body weight and breast cancer risk showing that adult weight gain is a better predictor of breast cancer risk than absolute weight or BMI.[8–10 29] Increases in waist circumference have also been reported to increase risk of other cancers such as pancreatic,[34] endometrial[35] and ovarian[36] cancer. A more equivocal issue in the literature is whether weight loss has a protective effect. Our findings that a decrease in SS is associated with decreased risk are in line with the studies showing reduction of weight can reduce risk of postmenopausal breast cancer.[8 11]

Current SS (at postmenopausal age; HR=1.05, 95% CI 1.01 to 1.09), but not SS in the 20 s, was associated with a statistically significant increase in breast cancer risk and appears to be a better predictor compared to BMI. Our findings come in agreement with other studies[29] and in keeping with the findings of a comparison of adiposity measures in postmenopausal women where waist circumference was found to be a stronger risk factor than BMI for a variety of conditions.[37] Absolute waist circumference has also been associated with increased risk of other cancers such as lung[38] and colon.[39]

In our study changes in SS was a better predictor of breast cancer risk compared to absolute SS in terms of strength of association. Although the exact mechanism of these relationships need to be better understood, there is a suggestion that body fat around the waist is more metabolically active than adipose tissue elsewhere.[40] Obesity is known to increase oestrogen levels as a result of aromatisation of androstenedione in adipose tissue as well as affect insulin resistance and chronic inflammation, well known factors shown to increase breast cancer risk.[41]

## CONCLUSION

Convincing evidence is available that high-body weight, BMI and adult weight gain, which typically reflect an increase in body fat, are associated with increased breast cancer risk. To the best of our knowledge, this is the first study investigating the association of waist changes using SS as a proxy and breast cancer risk. Between 20s and postmenopausal age, an increase in skirt size by one unit every decade increases the risk of postmenopausal breast cancer by 33%. Validation of these results could provide women with a simple and easy to understand message.

**Author affiliations**
[1]Department of Women's Cancer, Institute for Women's Health, University College London, London, UK
[2]University College London Hospitals NHS Foundation Trust, London, UK
[3]MRC Clinical Trials Unit, London, UK
[4]Academic Health Science Centre, University of Manchester, Manchester, UK

**Acknowledgements** The authors are particularly grateful to the women throughout the UK who are participating in the trial and to the entire medical, nursing and administrative staff who work on UKCTOCS.

**Contributors** UM, E-OF and MB designed the study. E-OF, CC, LCS and UM undertook the literature research. E-OF, MB and UM drafted the manuscript. MB and E-OF prepared the figures and tables and MB undertook the statistical analysis. All authors were involved in data collection. E-OF, CC and LCS were

involved in data cleaning. All authors contributed to interpretation of the data and critically revised the manuscript and approved the final version.

**Funding** UKCTOCS was core funded by the Medical Research Council, Cancer Research UK and National Institute of Health Research (NIHR) with additional support from the Eve Appeal and the NIHR Biomedical Research Centre at UCLH/UCL. The views expressed in the publication are those of the authors and not necessarily those of the funders. The funding source or the sponsor had no role in data collection, data analysis, data interpretation or writing of the report. The researchers are independent from the funders.

**Competing interests** UM and IJ have a financial interest through Abcodia Ltd in the third party exploitation of clinical trials biobanks, which have been developed through the research at UCL. During part of the trial IJ had a consultancy arrangement with Becton Dickinson in the field of tumour markers.

**Patient consent** Obtained.

**Ethics approval** The study was approved by the Joint University College London/University College London Hospital Committees on the Ethics of Human Research (REC reference: 06/Q0505/102, June 2008). Approvals for the use of hospital episode statistics data were obtained as part of the standard hospitals episode statistics approval process.

**Provenance and peer review** Not commissioned; externally peer reviewed.

**Data sharing statement** No additional data are available.

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
