## [Reviewer comments · BMJ Open]

Some articles will have been accepted based in part or entirely on reviews undertaken for other BMJ Group journals. These will be reproduced where possible.

ARTICLE DETAILS

TITLE (PROVISIONAL)	Association of skirt size and postmenopausal breast cancer risk in older women: a cohort study within the UK Collaborative Trial of Ovarian Cancer Screening (UKCTOCS)
AUTHORS	Fourkala, Evangelia-Ourania; Burnell, Matthew; Cox, Catharine; Ryan, Andy; Salter, Laura; Gentry-Maharaj, Aleksandra; Parmar, Mahesh; Jacobs, Ian; Menon, Usha

VERSION 1 - REVIEW

REVIEWER	Dr Mona Jeffreys Senior Lecturer in Epidemiology School of Social and Community Medicine University of Bristol UK
REVIEW RETURNED	15-Apr-2014

GENERAL COMMENTS	The authors need to reconsider the treatment of their exposure variables. Most of the variables treated as continuous variables would be better off categorised. For example, - a unit change in IMD score is meaningless.- For Change in SS it is clearly not appropriate to use mean (SD) as the SD>mean, implying skewness. Please report medians for all skewed variables.- reporting a HR per 100,000 cigarettes (equates to per 13 years of 20 cigs/day) is v unusual, please use more standard categorisations- at least for the main exposures (SS and CSS), it is vital to a reader to know how these are distributed, so presenting as a categorical variable would be more helpful. In Table 1B, some categories should be collapsed for meaningful interpretation, e.g. Daughter BC history should be 0 or 1+. If a one unit increase in SS is from e.g. size 10 to 12, it is unclear what a 0.5 unit increase (see suppl tables) means, since most clothes do not come in odd sizes. In the multiple imputation model, it appears that outcome was not included, which may lead to biased estimates. Please could the authors either clarify, or re-run. For each table it should be specified whether the results presented are derived from the complete case analysis or MI analysis, and for the former, the number of people with missing data in each category of exposure variable should be reported. There is no need to include SE in tables in which HR and CI are presented. Note that P cannot = 0.000 or 1.0. Making these changes
--

	will make the tables more readable. Page 12: it is not appropriate to use P values to compare the strength of the associations between various exposures. This paragraph should be re-written, focussing on the magnitude of the HRs. The first sentence on Page 14 needs to be re-written for a non-statistician (removing the word dbetas). If the frac poly analysis showed that the linear assumption was best, surely this can be removed from the manuscript, and summarised in one sentence. Recall bias cannot occur in cohort studies. This should be omitted. The authors statement regarding non-differential misclassification is correct. No mention is made of the fact that misclassification might be different across the SS continuum, i.e. reporting of size 10 and 12 might be more accurate than reporting of size 20 and 22. The effect of this limitation needs to be considered. The authors statement about "modern downsizing of SS" is important, and deserves more attention. If this is true (I have heard anecdotally, but do not have evidence of this), it has implications for the authors' conclusions. Regarding the issue of weight loss, it would be important to introduce a lag period into the study, to ensure that weight loss due to disease is not included. Since this was not done, the effect of weight loss may be greater than estimated here. It would be of interest to see how the outcomes from the various sources were concordant (or discordant) with each other. Might there be non-differential misclassification of outcome? The STROBE checklist included is just the checklist; the authors are required to indicate on what page each item has been included, but this does not appear to have been done. Figures S1 and S2 are probably not needed, as they do not add to the interpretation of the data.
--	--

REVIEWER	Emily White University of Washington and Fred Hutchinson Cancer Research Center, Seattle, WA, USA
REVIEW RETURNED	25-Apr-2014

GENERAL COMMENTS	Interesting study, but description of methods and statistical approach need revision. This paper presents the first study of change in waist circumference over adult life in relation to breast cancer risk. It is based on a well conducted cohort study, with innovative exposure measurement of waist circumference (recall of skirt size (SS)) and it has interesting findings. My main concerns with the paper are that the paper is not clearly written and the data are not analyzed as one expects in an epidemiologic study. Specifically:
---

1. The terms recruitment, baseline, follow-up questionnaire and current-study entry, are confusing. I think only two time points are of interest, so I'd stick to the terms recruitment and current-study entry throughout manuscript and tables.

2. Abstract "SS at baseline (sixties)"—remove sixties—it is confusing.

3. Introduction—It defies common sense to say that clothing size may be independent of BMI—plus the correlation between SS and BMI in this study is .64.

4. Methods section needs to be reorganized. It should be: Setting (describe the trial, recruitment and current study entry, but not follow-up) , Subjects (explain who entered this study, who was excluded, ie all women with breast cancer prior to current-study entry, right? and numbers of subjects before and after exclusion), Exposure data collection (measurement of height, weight and skirt size, calculation of change in skirt size, other covariates), Follow-up for breast cancer and censoring (give methods of follow-up and dates of follow-up for these endpoints —from current study entry until July 1 , 2010? Do not give dates the file received --that is extraneous), Statistical methods (see below).

5. Make clear whether height and weight were self reported or measured.

6. Make clear what HES notification is.

7. The statistical methods section is poorly structured and confusing. Specifically:

a. The first 6 sentences I think refer to calculation of change in SS and should be under exposure measurement.

b. The next 4 sentences make no sense and should be dropped. This section says that women were considered at risk from the age of 25, with delayed entry (left truncation to study entry), then says this " takes into account that a woman's contribution to the analysis is conditional on having survived (from developing BC for the first time) until date of questionnaire". But it then says that women with BC previous to study-entry questionnaire completion date were excluded from the analysis. If women with prior BC were excluded, as they should be, the prior sentences make no sense. At risk time simply begins at study entry. Also see comment 8.

c. Then it continues "Equating the date for a given age (here, when age=25) with onset of risk means that the analysis is already age-adjusted so that there is no need for age as an additional covariate." This is not correct. If calendar time (date) is the timeline in the Cox regression, only calendar time is controlled for as a confounder, not age. See comment 8.

d. The next (2nd) paragraph need to be simplified and put under the section follow-up for breast cancer and censoring.

e. The 3rd paragraph. The section on the grouping of CSS into 4 groups should be dropped—CSS is treated as continuous in this paper. (The 4 groups seem to be only used in the supplemental tables, and it is clear enough when one looks at those tables.) This paragraph should explain the statistical methods used in the tables in the paper, and how covariates to include in the models were determined.

f. Paragraph 4 could be combined with 3. For the part on functional form of CSS, you should say you evaluated the best functional form and it was determined to be linear CSS with the (log) relative hazard. (As it stands now, you state that the form for CSS is polynomial in the methods, then in results you present it as linear, and then in discussion you say linear was the best form—hard for reader to comprehend in that order).

g. Paragraph 5—OK.

	8. The main statistical method needs to be presented more clearly. It was Cox model with entry (left truncation) at date (or age?) of study-entry questionnaire, and exit at earliest date (or age?) of breast cancer, death, loss to follow-up or end of follow-up (July, 1, 2010t?), right? Make clear if the Cox timeline was date or age. If it was date, then age MUST be a covariate in the Cox model. 9. Drop table 4, Figure 1 and 2 as these are not needed to understand the results of the study. Also, drop the description of these from Methods and Results. 10. Drop table S3, and just mention you imputed for missingness in the statistical methods section and in results. Also drop Supplemental figures S1 and S2 and just mention you checked the residuals in the statistical methods. 11. Table 1A, 1B, 3 and S2—drop standard error –this is not usually given and is redundant with 95% CI. 12. Table 1A—present BMI HR as per 5 kg/m² of BMI, otherwise it is too hard to interpret such a small HR. Also add number of women and number of breast cancer cases to title. 13. Table 1B—where is ref group for education? Does pill use mean oral contraceptive use—make clearer. 14. Table 1B and covariates in Table 3. Make a combined family history variable, eg number of first degree relatives with BC family history and use in Table 1B and to select for covariates. Now, only mother BC entered the model, while it seems as if a combined family history variable would have a stronger effect. 15. Tables S1A and S1B—keep as supplemental tables but mention these results briefly in Results. 16. The approach of selection of covariates for the model in Table 3 is not a conventional approach in epidemiology. In particular I am concerned that neither education nor smoking entered by your method of selecting covariates (potential or actual confounders), yet both education and smoking were associated with both BC risk and SS in this study. I suggest including all risk factors for BC in the model in Table 3 (all potential confounders) or including those factors that changed the beta for CSS by 10% (either when individually added to a model with only age and CSS, or using backward elimination from a model with CSS (but not SS at age 25, SS at entry or BMI) and all BC all risk factors , by first excluding the covariate that least changed the beta for CSS if that covariate was removed, then the second least change etc.) 17. Table 3. This should present the results for your main 4 exposures (SS at age 20, SS at study entry, CSS and BMI per 5 5 kg/m²), not for all factors in the final model. It is the main exposures that the reader is interested in. I would have two models –one adjusted for the covariates selected, then add a model with those covariates plus BMI. Footnote the covariates in the two models. Then Table S2 would be included in Table 3 and S2 could be dropped –it is table S2 that is of most interest in this paper.
--	---

VERSION 1 – AUTHOR RESPONSE

Reviewer Name: Dr Mona Jeffreys

The authors need to reconsider the treatment of their exposure variables. Most of the variables treated as continuous variables would be better off categorised. For example, a unit change in IMD score is meaningless.

We agree with the reviewer’s comment and we have changed IMD to standardised IMD, so that a unit

change reflects a change in 1 standard deviation, but we have retained it as continuous. Pregnancy less and more than 6 months has also been changed to categorical variable.

For Change in SS it is clearly not appropriate to use mean (SD) as the $SD > \text{mean}$, implying skewness. Please report medians for all skewed variables.

The CSS variable is not skewed especially, please see the added kernel density plot in Figure 1. It needs to be noted that the CSS variable can be also negative.

Reporting a HR per 100,000 cigarettes (equates to per 13 years of 20 cigs/day) is v unusual, please use more standard categorisations

We have changed smoking volume to a categorical variable ($0=0, 0<1 \leq 250, 250<2 \leq 500, 3>500$ where numbers are in 100 000s). We have kept the metric in total volume smoked (in 100 000s), even though it is not a common method. We have collected the data on number of cigarettes smoked a day and the period of smoking time. We feel that representing those variables in a combined manner is a more honest approach than the mean rate over a lifetime which may have no reflection on the reality of how those cigarettes were smoked. Units of 100 000s therefore becomes a natural unit to use given the actual numbers smoked.

At least for the main exposures (SS and CSS), it is vital to a reader to know how these are distributed, so presenting as a categorical variable would be more helpful.

We had presented both median and 25th and 75th percentiles for all variables as well as means and SDs. However for the 4 'adiposity' measures (SS in 20s, SS now, change in SS per time and BMI) have now been displayed graphically (with kernel density plots (Figure 1)) which we feel is the best way of all to appreciate the distributions.

In Table 1B, some categories should be collapsed for meaningful interpretation, e.g. Daughter BC history should be 0 or 1+.

We agree with the reviewer, we have reduced all cancer history for relatives into just 2 variables: relative with breast cancer (yes/no); relative with ovarian cancer (yes/no), where relative includes mother, grandmother, sister, daughter and aunt.

If a one unit increase in SS is from e.g. size 10 to 12, it is unclear what a 0.5 unit increase (see suppl tables) means, since most clothes do not come in odd sizes.

A 0.5 unit increase here is per 10 years, therefore it means a unit increase over 20 years. This supplementary table has now been excluded anyway, though the figure that displays this information has been retained as Figure 2.

In the multiple imputation model, it appears that outcome was not included, which may lead to biased estimates. Please could the authors either clarify, or re-run. For each table it should be specified whether the results presented are derived from the complete case analysis or MI analysis, and for the former, the number of people with missing data in each category of exposure variable should be reported.

The multiple imputation model has been rerun as suggested with all confounders included (as per the main model now). It has been also clarified in the methods sections that outcome (both time to event and event itself) has been included. Given that we are now presenting output only for the adiposity measure(s) (please see Table 3) by following the suggestion of the second reviewer, there is no place

in the output table to present missing data counts for the confounders. However missing data information is given in the multiple imputation section.

There is no need to include SE in tables in which HR and CI are presented. Note that P cannot = 0.000 or 1.0. Making these changes will make the tables more readable.

Standard error has been deleted as per reviewers comment.

Page 12: it is not appropriate to use P values to compare the strength of the associations between various exposures. This paragraph should be re-written, focussing on the magnitude of the HRs.

Unfortunately we do not know which section this reviewer is referring to - there is nothing of this on page 12 or neighbouring pages. Due to the new analysis though most of the results section has been re-written.

The first sentence on Page 14 needs to be re-written for a non-statistician (removing the word dfbetas). If the frac poly analysis showed that the linear assumption was best, surely this can be removed from the manuscript, and summarised in one sentence.

The definition of Dfbetas has now been excluded and this section has been re-written to hopefully make it clearer.

Recall bias cannot occur in cohort studies. This should be omitted. The author's statement regarding non-differential misclassification is correct.

It is possible to have recall bias in a retrospective cohort study. There is an issue of potential recall bias here in the main exposure variable (skirt size in their twenties) and we have now made it clear that this is a retrospective cohort study.

No mention is made of the fact that misclassification might be different across the SS continuum, i.e. reporting of size 10 and 12 might be more accurate than reporting of size 20 and 22. The effect of this limitation needs to be considered.

This may be the case and has now been mentioned as a possibility in the discussion, though we feel the issue is of minor consideration compared to the issue of misclassification itself.

The author's statement about "modern downsizing of SS" is important, and deserves more attention. If this is true (I have heard anecdotally, but do not have evidence of this), it has implications for the authors' conclusions.

We haven't found a reference to support our hypothesis of "modern downsizing of SS" but we felt that it something that should be considered in our discussion. We agree that if this is true it will have implications in our conclusions hence we have assumed that this downsizing trend does not noticeably occur over the period of 'current SS'.

Regarding the issue of weight loss, it would be important to introduce a lag period into the study, to ensure that weight loss due to disease is not included. Since this was not done, the effect of weight loss may be greater than estimated here.

We agree with the reviewer and we have included an additional analysis with a lag, where time of study entry is counted as 1 year past questionnaire date. This was the maximum time we could use to do the lag analysis since the follow-up time was relatively short, a limitation of the study that has been

mentioned in the manuscript.

It would be of interest to see how the outcomes from the various sources were concordant (or discordant) with each other. Might there be non-differential misclassification of outcome?

Similar analyses has been performed in the same cohort, please see references; Fourkala et al, Histological confirmation of breast cancer registration and self-reporting in England and Wales: a cohort study within the UK Collaborative Trial of Ovarian Cancer Screening. *BJC*, 2012. *Br J Cancer*. 2012 Jun 5;106(12):1910-6 and Gentry-Maharaj, Fourkala et al, Concordance of National Cancer Registration with self-reported breast, bowel and lung cancer in England and Wales: a prospective cohort study within the UK Collaborative Trial of Ovarian Cancer Screening. *Br J Cancer*. 2013 Nov 26; 109(11):2875-9. Overall, there is a good concordance between the different sources used to identify the breast cancer cases and this statement has now been added to the manuscript.

The STROBE checklist included is just the checklist; the authors are required to indicate on what page each item has been included, but this does not appear to have been done.

We have now added that.

Figures S1 and S2 are probably not needed, as they do not add to the interpretation of the data.

Figure S2 has been dropped, but we have retained Figure S1. Unless there is pressure on the amount of supplementary material that can be provided we feel that inclusion of such a plot, in addition to being good statistical practise, is in fact informative as it shows how the variable's effect changes with study time (or not, particularly, in this case - as is required by the PH model assumption).

Reviewer Name: Emily White

Interesting study, but description of methods and statistical approach need revision.

This paper presents the first study of change in waist circumference over adult life in relation to breast cancer risk. It is based on a well conducted cohort study, with innovative exposure measurement of waist circumference (recall of skirt size (SS) and it has interesting findings. My main concerns with the paper are that the paper is not clearly written and the data are not analyzed as one expects in an epidemiologic study.

Specifically:

1. The terms recruitment, baseline, follow-up questionnaire and current-study entry, are confusing. I think only two time points are of interest, so I'd stick to the terms recruitment and current-study entry throughout manuscript and tables.

We agree and it has been changed. We now refer just to recruitment and current study-entry.

2. Abstract "SS at baseline (sixties)"—remove sixties—it is confusing.

We agree, this has now been removed.

3. Introduction—It defies common sense to say that clothing size may be independent of BMI—plus the correlation between SS and BMI in this study is .64.

This has been corrected to the intended meaning, that SS could provide explanatory potential independently of BMI (additional information that BMI does not provide).

4. Methods section needs to be reorganized. It should be: Setting (describe the trial, recruitment and

current study entry, but not follow-up), Subjects (explain who entered this study, who was excluded, ie all women with breast cancer prior to current-study entry, right? and numbers of subjects before and after exclusion), Exposure data collection (measurement of height, weight and skirt size, calculation of change in skirt size, other covariates), Follow-up for breast cancer and censoring (give methods of follow-up and dates of follow-up for these endpoints —from current study entry until July 1, 2010? Do not give dates the file received -- that is extraneous), Statistical methods (see below).

Methods have been reorganized partially along the lines suggested but certain aspects were retained as we feel it is more appropriate. The date of files received is included to explain why the censorship point is not more recent. There is a necessity of the time buffer to be taken into account in order to make sure all events in the defined analysis period are known at the time of analysis.

5. Make clear whether height and weight were self-reported or measured.

We agree and we have now specified them as self-reported measures.

6. Make clear what HES notification is.

HES is Hospital of Episode statistics and it has been provided in the manuscript.

7. The statistical methods section is poorly structured and confusing. Specifically:

a. The first 6 sentences I think refer to calculation of change in SS and should be under exposure measurement.

b. The next 4 sentences make no sense and should be dropped. This section says that women were considered at risk from the age of 25, with delayed entry (left truncation to study entry), then says this “takes into account that a woman’s contribution to the analysis is conditional on having survived (from developing BC for the first time) until date of questionnaire”. But it then says that women with BC previous to study-entry questionnaire completion date were excluded from the analysis. If women with prior BC were excluded, as they should be, the prior sentences make no sense. At risk time simply begins at study entry. Also see comment 8.

This section has been changed to hopefully make it clearer. In response to the issues mentioned, left truncation means that the subject was exposed to risk prior to the study but was unobserved during that time. We know a posteriori that they didn’t die (or get BC in our example) because we observe them in the study, but there are others who never get to be ‘observed’ because they died beforehand (and in our example, it is not just those we excluded by design who had a prior BC – there will be those who never joined our trial because of their BC, certainly if they died from it). Therefore we need to account for the fact that had they failed we would never have encountered the subject.

c. Then it continues “Equating the date for a given age (here, when age=25) with onset of risk means that the analysis is already age-adjusted so that there is no need for age as an additional covariate.” This is not correct. If calendar time (date) is the timeline in the Cox regression, only calendar time is controlled for as a confounder, not age. See comment 8.

Age is the time metric used for the analysis (so is controlled for in the analysis). Technically the origin for each woman is age=25 rather than age=0 to reflect the SS question, though mathematically it will make no difference. The mentioning of date was merely to show how each woman was lined up with age=25 as the same starting point (obviously by using the date this happened). We have tried to make this section clearer.

d. The next (2nd) paragraph need to be simplified and put under the section follow-up for breast cancer and censoring.

This has been moved to general methods.

e. The 3rd paragraph. The section on the grouping of CSS into 4 groups should be dropped—CSS is treated as continuous in this paper. (The 4 groups seem to be only used in the supplemental tables, and it is clear enough when one looks at those tables.) This paragraph should explain the statistical methods used in the tables in the paper, and how covariates to include in the models were determined.

We have dropped the whole section, including supplemental tables, concerning categorisation of CSS and its cross-tabulation with the other confounders.

f. Paragraph 4 could be combined with 3. For the part on functional form of CSS, you should say you evaluated the best functional form and it was determined to be linear CSS with the (log) relative hazard. (As it stands now, you state that the form for CSS is polynomial in the methods, then in results you present it as linear, and then in discussion you say linear was the best form—hard for reader to comprehend in that order).

We need to clarify that in the methods we had stated that we looked at the most appropriate functional form from a pre-defined (and standard) set of possible powers, of which linear (power=1) was one possibility. The section on statistics has been re-written hopefully making it clearer how the analysis was performed.

g. Paragraph 5—OK.

8. The main statistical method needs to be presented more clearly. It was Cox model with entry (left truncation) at date (or age?) of study-entry questionnaire, and exit at earliest date (or age?) of breast cancer, death, loss to follow-up or end of follow-up (July, 1, 2010t?), right? Make clear if the Cox timeline was date or age. If it was date, then age MUST be a covariate in the Cox model.

As mentioned before we hope that this should be clearer now – to summarise it is: origin (age 25), date of entry (age at questionnaire), date of exit (breast cancer or July 1st 2010), time metric is age (minus 25).

9. Drop table 4, Figure 1 and 2 as these are not needed to understand the results of the study. Also, drop the description of these from Methods and Results.

Table 4 has been dropped and Figure 2 has been moved to supplementary figures. Figure 1 has been retained (as Figure 2) as we feel it is helpful to show the effects on the HR of CSS values other than a unit change, especially as this is in the hazard metric and not log-hazard. Additionally, the inclusion of confidence bands further helps with judging the degree of risk change with differing exposure changes – the ‘dose-response’ relationship.

10. Drop table S3, and just mention you imputed for missingness in the statistical methods section and in results. Also drop Supplemental figures S1 and S2 and just mention you checked the residuals in the statistical methods.

Table 3 has been dropped although the main line (for CSS) has been incorporated into the main table that has (adjusted) HRs now for only the adiposity measures in the various models presented. Supplementary figure 1 has been kept (now Supplementary figure 2) as per response to the first reviewer’s comment and Supplementary figure 2 has been dropped.

11. Table 1A, 1B, 3 and S2—drop standard error –this is not usually given and is redundant with 95%

CI.

Standard error has been deleted as per reviewers comment.

12. Table 1A—present BMI HR as per 5 kg/m² of BMI, otherwise it is too hard to interpret such a small HR. Also add number of women and number of breast cancer cases to title.

We have changed all the HRs pertaining to BMI, as reflective of a 5 unit (kg/m²) change. We have included the values for n and events for each model in the table now as they change (slightly) between the models.

13. Table 1B—where is ref group for education? Does pill use mean oral contraceptive use—make clearer.

Each education group should in fact have its own reference group, as the education variables are not at all mutually exclusive (and they are not readily put into a natural order), and each has been included now. We have also made it clear that pill use is oral contraceptive pill use.

14. Table 1B and covariates in Table 3. Make a combined family history variable, e.g. number of first degree relatives with BC family history and use in Table 1B and to select for covariates. Now, only mother BC entered the model, while it seems as if a combined family history variable would have a stronger effect.

As with response to the first reviewer, we have reduced all cancer history for relatives into just 2 variables: relative with breast cancer (yes/no); relative with ovarian cancer (yes/no), where relative includes mother, grandmother, sister, daughter and aunt.

15. Tables S1A and S1B—keep as supplemental tables but mention these results briefly in Results.

We have actually removed tables S1A and S1B.

16. The approach of selection of covariates for the model in Table 3 is not a conventional approach in epidemiology. In particular I am concerned that neither education nor smoking entered by your method of selecting covariates (potential or actual confounders), yet both education and smoking were associated with both BC risk and SS in this study. I suggest including all risk factors for BC in the model in Table 3 (all potential confounders) or including those factors that changed the beta for CSS by 10% (either when individually added to a model with only age and CSS, or using backward elimination from a model with CSS (but not SS at age 25, SS at entry or BMI) and all BC all risk factors by first excluding the covariate that least changed the beta for CSS if that covariate was removed, then the second least change etc.)

We agree with the reviewer and as stated in the beginning of this response we have re-analysed the data along the lines suggested in comments 16 and 17.

17. Table 3. This should present the results for your main 4 exposures (SS at age 20, SS at study entry, CSS and BMI per 5 kg/m²), not for all factors in the final model. It is the main exposures that the reader is interested in. I would have two models –one adjusted for the covariates selected, then add a model with those covariates plus BMI. Footnote the covariates in the two models. Then Table S2 would be included in Table 3 and S2 could be dropped –it is table S2 that is of most interest in this paper.

We agree with the reviewer and as stated in the beginning of this response we have re-analysed the data along the lines suggested in comments 16 and 17.

VERSION 2 – REVIEW

REVIEWER	Peter Baade Cancer Council Queensland Australia
REVIEW RETURNED	18-Jul-2014

GENERAL COMMENTS	This was a very interesting paper to read, and well written and analysed. I found the description of the time periods in the methods confusing, and only started to make sense once reading the results. Can this be clarified in the methods? Also, using “recruitment questionnaire” and “study entry questionnaire” to describe the two survey periods is confusing, since they could be taken to convey the same meaning. For what period were breast cancer diagnoses ascertained? I assume it was just the period following the second questionnaire, which is the entry date for this study. Given this study is considering incidence risk, some introductory comment explaining why you are using standard survival analysis methods (page 7, line 51) may be beneficial for clarity. The main limitation to these results is that, as identified by the authors, skirt size measurements have changed over the year. While the authors have provided a reasonable discussion of the potential impact of this, does this have implications for the ongoing assessment of skirt sizes? That is, is there potential for a false sense of security (ie. lower risk) if the lack of change in skirt size is due to manufacturer specifications rather than effective weight control strategies? It would be useful have more information about the age distribution of the cohort, instead of just 75% were aged over 60 years of age. This would help to better understand the potential for recall bias, particularly given the prospective component of the study was relatively very small. The correlation between the different adiposity measures is high, as acknowledged by the authors. Was any data available about change in BMI? As it stands, the measure relating to change in skirt size is the only weight gain measure assessed. Abstract – It appears that the hazard ratios presented here are from Table 3, in which case they are not univariable models, but from the model adjusted for potential confounders. Finally, I thought the number of abbreviations throughout the text made it difficult to follow.
--

VERSION 2 – AUTHOR RESPONSE

This was a very interesting paper to read, and well written and analysed.

I found the description of the time periods in the methods confusing, and only started to make sense once reading the results. Can this be clarified in the methods?

We are not sure on how to make this clearer as we did try in the first round of revisions to clarify this part as much as possible. If the reviewer could tell us what is confusing it may be easier to make the changes.

Also, using “recruitment questionnaire” and “study entry questionnaire” to describe the two survey periods is confusing, since they could be taken to convey the same meaning.

These are two different questionnaires. All subjects completed a questionnaire at recruitment and a further postal follow-up questionnaire 3-4 years post-recruitment. The latter was the study-entry questionnaire for the current analysis.

For what period were breast cancer diagnoses ascertained? I assume it was just the period following the second questionnaire, which is the entry date for this study.

Correct, the breast cancer cases were ascertained for the period following the second questionnaire.

Given this study is considering incidence risk, some introductory comment explaining why you are using standard survival analysis methods (page 7, line 51) may be beneficial for clarity.

We are considering the association between CSS and BC risk – we are not trying to model the underlying incidence rate itself. The word incidence has been used in the text to distinguish it from mortality. By using a Cox model we are avoiding the specification of an underlying baseline risk level (which is not the case when we use the exponential regression model to estimate absolute risks). We have added to the text ‘Standard survival analysis methods were used to analyse the data.’ ‘Cox models were preferred so that specification of an underlying hazard was not required’.

The main limitation to these results is that, as identified by the authors, skirt size measurements have changed over the year. While the authors have provided a reasonable discussion of the potential impact of this, does this have implications for the ongoing assessment of skirt sizes? That is, is there potential for a false sense of security (ie. lower risk) if the lack of change in skirt size is due to manufacturer specifications rather than effective weight control strategies?

This is an interesting point of the reviewer. We have shown that increasing SS is associated with increased breast cancer risk so whether people underestimate this risk increases or not in the future, doesn't alter our conclusion. By using a more objective measure like waist size (which presumably is highly correlated with skirt size) can track any actual changes.

It would be useful have more information about the age distribution of the cohort, instead of just 75% were aged over 60 years of age. This would help to better understand the potential for recall bias, particularly given the prospective component of the study was relatively very small.

We agree with the reviewer. We have done a kernel density plot for age at study entry questionnaire and we have included this information as supplementary material.

The correlation between the different adiposity measures is high, as acknowledged by the authors. Was any data available about change in BMI? As it stands, the measure relating to change in skirt size is the only weight gain measure assessed.

We only have an historic estimate for SS so this is the only change measure we have. This is one of the points that we believe that skirt size is a good measure of adiposity because people might also be able to recall it better from their youth than waist size or weight (and height).

Abstract – It appears that the hazard ratios presented here are from Table 3, in which case they are not univariable models, but from the model adjusted for potential confounders.

We have corrected that.

Finally, I thought the number of abbreviations throughout the text made it difficult to follow. We have changed some of the abbreviations to full text to make it easier for the reader.